



# Mapping the global distribution of lead and its isotopes in seawater with explainable machine learning

Arianna Olivelli[1,2], Rossella Arcucci[1], Mark Rehkämper[1], Tina van de Flierdt[1]

[1] Grantham Institute for Climate Change and the Environment, Imperial College London, South Kensington Campus, London SW7 2AZ, United Kingdom
[2] Department of Earth Science and Engineering, Imperial College London, South Kensington Campus, London SW7 2AZ, United Kingdom

*Correspondence to*: Arianna Olivelli (a.olivelli21@imperial.ac.uk)

**Abstract.** Lead (Pb) and its isotopes are a powerful tool to study the pathways of Pb pollution from land to sea and, simultaneously, investigate biogeochemical processes in the ocean. However, the scarcity and sparsity of *in situ* measurements of Pb concentrations and isotope compositions do not allow for a comprehensive understanding of Pb pollution pathways and biogeochemical cycling on a global scale. Here, we present three machine learning models developed to map seawater Pb concentrations and isotope compositions leveraging the global GEOTRACES dataset as well as historical data. The models use climatologies of oceanographic and atmospheric variables as features from which to predict Pb concentrations, $^{206}Pb/^{207}Pb$, and $^{208}Pb/^{207}Pb$. Using Shapley Additive Values (SHAP), we found that seawater temperature, atmospheric dust and black carbon, and salinity are the most important features for predicting Pb concentrations. Dissolved oxygen concentration, salinity, temperature, and atmospheric dust are the most important features for predicting $^{206}Pb/^{207}Pb$, while atmospheric black carbon and dust, seawater temperature, and surface chlorophyll-a for $^{208}Pb/^{207}Pb$. Our model outputs show that (i) the surface Indian Ocean has the highest levels of pollution, (ii) pollution from previous decades is sinking in the North Atlantic and Pacific Ocean, and (iii) waters characterised by a highly anthropogenic Pb isotope fingerprint are spreading from the Southern Ocean throughout the Southern Hemisphere at intermediate depths. By analysing the uncertainty associated with our maps, we identified the Southern Ocean as the key area to prioritise in future sampling campaigns. Our datasets, models and their outputs, in the form of Pb concentration, $^{206}Pb/^{207}Pb$, and $^{208}Pb/^{207}Pb$ climatologies, are made freely available to the community at Olivelli et al. (2024a, https://doi.org/10.5281/zenodo.14261154) and https://github.com/OlivelliAri/Pb-ML_GEOTRACES.

## 1 Introduction

The low concentrations at which lead (Pb) is present in the ocean (on the order of parts per trillion) and the ubiquity of Pb contamination during sampling and sample processing did not allow for successful measurements of seawater Pb concentration until 1963 (Tatsumoto & Patterson, 1963) and isotope composition until 1976 (Schaule & Patterson, 1981). Indeed, since the early 1900s, human activities have caused an increase in Pb concentrations in the environment, both on land and in the ocean, which altered the background levels and the biogeochemical cycling of Pb (Boyle et al., 2014; Nriagu, 1979; Nriagu & Pacyna, 1988; Pacyna & Pacyna, 2001).

This increase in Pb concentrations has been accompanied by a shift in the Pb isotope compositions of seawater, reflecting the predominance of anthropogenically-sourced over naturally-sourced Pb (Boyle et al., 2014). In fact, the relative abundances of stable Pb isotopes ($^{204}$Pb, $^{206}$Pb, $^{207}$Pb, and $^{208}$Pb), expressed as ratios such as $^{206}$Pb/$^{207}$Pb and $^{208}$Pb/$^{207}$Pb, can be used to identify and quantify the contributions of the different sources of Pb (Reuer & Weiss, 2002).

In the years and decades that followed the first successful measurements of Pb and its isotopes in seawater, sampling efforts mainly concentrated in the North Atlantic Ocean (Alleman et al., 1999; Boyle et al., 1986; Helmers et al., 1991; Pohl et al., 1993; Shen & Boyle, 1988; Sherrell et al., 1992; Véron et al., 1994, 1998, 1999). This basin, surrounded by the early-developed economies of North America and Western Europe, initially saw a sharp increase in pollution followed by a decrease thanks to environmental policies that phased out and banned the use of leaded gasoline since the 1970s (Bridgestock et al., 2016; Weiss et al., 2003).

A major breakthrough for the understanding of the marine biogeochemistry and pollution of Pb on a global scale came with the international marine geochemistry programme GEOTRACES, which has run since 2006 and included Pb concentrations and isotope compositions as key parameters of interest to measure on all its cruise sections (GEOTRACES Planning Group, 2006). However, despite these great efforts, the majority of the world's ocean remains unsampled for Pb concentrations and isotope compositions, especially in the Southern Hemisphere (Fig. 1).

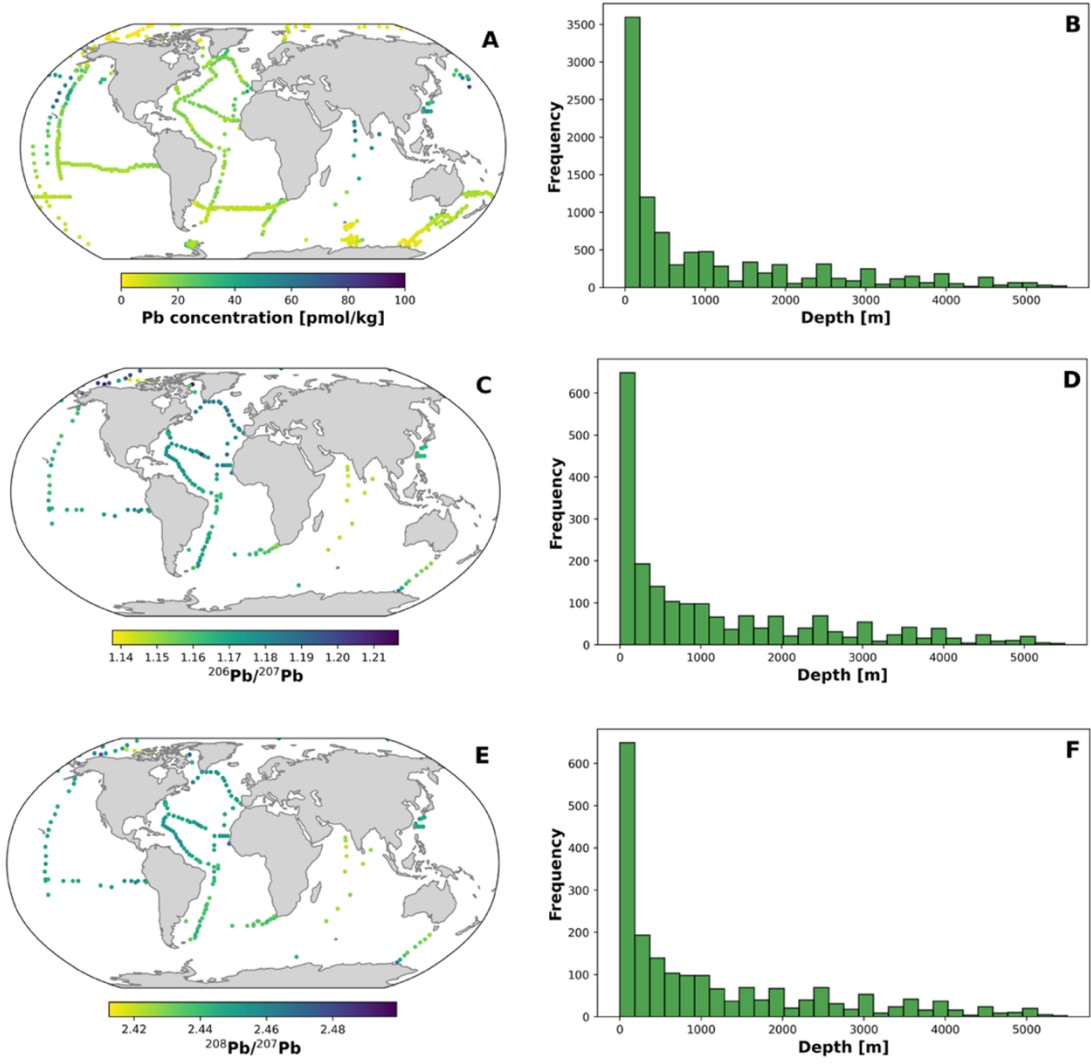

**Figure 1. Geographical distribution of all samples in the Pb concentration (A), [206]Pb/[207]Pb (C), and [208]Pb/[207]Pb (E) datasets. Panels A, C, and E include all samples from the top 100 m. Panels B, D, and F represent the frequency distribution of sampling depths for panels A, C, and E, respectively.**

In this context, modelling studies represent a powerful tool to expand our knowledge about the distribution and biogeochemical cycle of Pb, complementary to *in situ* observations. While deterministic models require a good understanding of the processes at play in the biogeochemical of Pb, machine learning (ML) models do not require such prior knowledge as they are data-driven (Glover et al., 2011). Therefore, the latter can be regarded as a computationally efficient way to investigate the

distribution of Pb and its isotopes in the global ocean without the need to explicitly parametrise causal relationships derived from fundamental knowledge of the biogeochemical cycle of Pb.

In recent years, ML algorithms have been employed to investigate several topics within chemical oceanography, including the distribution and cycling of tracers, such as iron (Huang et al., 2022), zinc (Roshan et al., 2018), copper (Roshan et al., 2020),

and barium (Mete et al., 2023), iodide (Sherwen et al., 2019), and nitrogen isotopes (Rafter et al., 2019). A thorough overview of the use and potential of ML in ocean sciences is provided in Sonnewald et al. (2021).

In this study, we leverage the growing GEOTRACES dataset, in combination with climatologies of several oceanographic and atmospheric variables, to map the distribution of Pb and its isotopes in the global ocean using the ML regression algorithm XGBoost. With this approach, we aim to (i) produce three-dimensional global maps of Pb concentrations and isotope

compositions ($^{206}$Pb/$^{207}$Pb and $^{208}$Pb/$^{207}$Pb) to use as a benchmark for future levels of pollution, and to (ii) identify areas where to concentrate sampling efforts in the upcoming years and decades by analysing the uncertainty of our model outputs. We make the resulting data products available to the community, in the form of mean global climatologies, and envisage updating the models' outputs as new observations become available.

## 2 Materials and Methods

Regression ML models require training on existing observations of a target variable (to be predicted by the model) alongside a number of features (i.e. ancillary variables) in order to make accurate predictions. Therefore, three different datasets were compiled to develop three separate ML models to map the global distribution of Pb concentrations, $^{206}$Pb/$^{207}$Pb, and $^{208}$Pb/$^{207}$Pb, respectively. Each dataset includes information on 14 geographical, oceanographic and atmospheric features (Table 1) and a target variable (either Pb concentration, $^{206}$Pb/$^{207}$Pb, or $^{208}$Pb/$^{207}$Pb). The features included in the datasets were chosen due to

their proven predictive power in other ML studies focusing on ocean chemistry (Huang et al., 2022; Mete et al., 2023; Roshan et al., 2018, 2020; Sherwen et al., 2019) and their likely connection to the biogeochemical cycle of Pb.

The three datasets and models were built following the same procedure, explained in sections 2.1 and 2.2. All data preparation, modelling and analyses were done in Python 3.11.5 in a Linux operating system.



**Table 1. List of features used for dataset compilation and model development for Pb concentration, $^{206}$Pb/$^{207}$Pb, and $^{208}$Pb/$^{207}$Pb.**

| Feature name | Data source | Spatial resolution | Temporal resolution |
|---|---|---|---|
| ***Geographical features*** | | | |
| *Latitude [°N]** | *Sampling information* | | |
| *Longitude [°E]** | *Sampling information* | | |
| Depth [m] | Sampling information | | |
| Distance from bottom depth [m] | Derived from sampling information | | |
| ***Oceanographic features*** | | | |
| Temperature [°C] | World Ocean Atlas 2018 | 1°× 1° | Annual climatology, 01/2005 - 12/2017 |
| Salinity | World Ocean Atlas 2018 | 1°× 1° | Annual climatology, 01/2005 - 12/2017 |
| Dissolved nitrate [µmol/kg] | World Ocean Atlas 2018 | 1°× 1° | Annual climatology, all available data (1955 - 2017) |
| Dissolved phosphate [µmol/kg] | World Ocean Atlas 2018 | 1°× 1° | Annual climatology, all available data (1955 - 2017) |
| Dissolved silicate [µmol/kg] | World Ocean Atlas 2018 | 1°× 1° | Annual climatology, all available data (1955 - 2017) |
| Density anomaly ($\sigma_0$) [kg/m$^3$] | World Ocean Atlas 2018 | 1°× 1° | Annual climatology, 01/2005 - 12/2017 |
| Dissolved oxygen [µmol/kg] | World Ocean Atlas 2018 | 1°× 1° | Annual climatology, all available data (1955 - 2017) |
| Apparent oxygen utilization [µmol/kg] | World Ocean Atlas 2018 | 1°× 1° | Annual climatology, all available data (1955 - 2017) |
| Mixed layer depth [m] | World Ocean Atlas 2018 | 1°× 1° | Annual climatology, 01/2005 - 12/2017 |
| Chlorophyll-a [mg/m$^3$] | CMEMS[a] | 0.25°× 0.25° | Monthly, 01/2005 - 12/2017 |
| ***Atmospheric features*** | | | |
| Black carbon AOD[b] | CAMS[c] | 0.75°× 0.75° | Monthly, 01/2005 - 12/2017 |
| Dust AOD[b] | CAMS[c] | 0.75°× 0.75° | Monthly, 01/2005 - 12/2017 |

[*] Variables used for dataset compilation and for an initial version of the models, but dropped for the final models (see more in SI note 1)

[a] Copernicus Marine Environment Monitoring Service

[b] Aerosol Optical Depth

[c] Copernicus Atmosphere Monitoring Service

## 2.1 Data sources

### 2.1.1 Pb concentrations and isotope compositions

Observations of seawater dissolved and total dissolvable Pb concentrations, $^{206}$Pb/$^{207}$Pb, and $^{208}$Pb/$^{207}$Pb included in the GEOTRACES Intermediate Data Product 2021v2 (IDP2021v2; GEOTRACES Intermediate Data Product Group, 2023) form the basis of this study. Only GEOTRACES IDP2021v2 data with quality control flag equal to 1, equivalent to good data, were included in our Pb concentrations, $^{206}$Pb/$^{207}$Pb, and $^{208}$Pb/$^{207}$Pb datasets. Among these good data points, 8 samples from the southern sector of the Indian Ocean and the Southern Ocean were not included in the Pb concentrations dataset as they had unusually high values compared to those collected at surrounding locations. Additionally, one sample from the Arctic Ocean was not included in our dataset as it had a negative Pb concentration (-0.05 pmol/kg).

Additional data were included in our dataset to increase the number of observations available; these encompass published (Chen et al., 2023a; Chien et al., 2017; Lanning et al., 2023; Olivelli et al., 2023, 2024b) and unpublished studies (from the

MAGIC Laboratories at Imperial College London). As previous studies have shown that dissolved and total dissolvable Pb concentrations and isotope compositions are comparable in the open ocean (Bridgestock et al., 2018; Olivelli et al., 2023, 2024b; Schlosser et al., 2019), data for dissolved and total dissolvable Pb concentrations, $^{206}Pb/^{207}Pb$, and $^{208}Pb/^{207}Pb$ were pooled together for the purpose of this work.

### 2.1.2 Oceanographic and atmospheric features

Global gridded climatologies of seawater temperature, salinity, density anomaly ($\sigma_0$), nitrate, phosphate, silicate, oxygen, apparent oxygen utilization (AOU), and mixed layer depth (MLD) were obtained from the World Ocean Atlas 2018 product (WOA18; Garcia, 2019). The WOA18 has a $1° \times 1°$ spatial resolution and 102 depth levels (spaced every 5 m between 0 and 100 m, every 25 m between 100 and 500 m, every 50 m between 500 and 2000 m, and every 100 m between 2000 and 5500 m). Chlorophyll-a concentrations in the surface ocean were obtained from the Copernicus Marine Service Global Ocean Biogeochemistry Hindcast product ($0.25° \times 0.25°$ spatial resolution) as monthly averages for the period between January 2005 and December 2017, and averaged over the entire period to create a climatology.

Dust aerosol optical depth (AOD) and black carbon AOD were included as most Pb enters the ocean via atmospheric transport. Dust was found to be an important source of natural Pb to the surface ocean in several regions, including the Atlantic Ocean (Bridgestock et al., 2016; Olivelli et al., 2023), southern Indian Ocean (Lee et al., 2015), and Red Sea (Benaltabet et al., 2020). Black carbon, on the other hand, was chosen as an indicator of Pb emissions from industrial and high-temperature activities, which are known sources of anthropogenic Pb to the environment (Nriagu & Pacyna, 1988; Pacyna & Pacyna, 2001). Data for the years between January 2005 and December 2017 were obtained from CAMS global reanalysis (ECMWF Atmospheric Composition Reanalysis 4; EAC4) monthly averaged fields, with a $0.75° \times 0.75°$ spatial resolution, and were averaged over the whole period to obtain a climatology.

### 2.1.3 Data handling and dataset preparation

The WOA18 grid was chosen as a reference for all feature and target variables in this study. Therefore, Pb concentrations, $^{206}Pb/^{207}Pb$, $^{208}Pb/^{207}Pb$, chlorophyll-a, dust AOD, and black carbon AOD values, which were either obtained as point observations or gridded at higher resolution, were re-gridded according to it. Lead data were binned to the $1° \times 1°$ grid cell the

samples were collected in and assigned to the depth level closest to the depth at which the samples were collected. All samples collected at depth larger than 5500 m were discarded, as 5500 m is the deepest depth level considered in the WOA18. Distance from bottom depth was calculated as the difference between the bottom depth value reported for the sampling station and the

130 WOA18 depth level to which the observation was assigned. Chlorophyll-a, dust AOD, and black carbon AOD were re-gridded by assigning each 1°× 1° grid cell the feature value of the respective product cell located the closest to the centre of the WOA18-gridded cell.

All samples in each 1°× 1° column were attributed the same dust AOD, black carbon AOD, surface chlorophyll-a, and mixed layer depth values, regardless of their depth. This was done to assess whether characteristics of the surface ocean have an

135 important impact throughout the water column. These features represent atmospheric inputs (dust AOD and black carbon AOD), surface primary production as a proxy for sinking particulate matter in the water column (surface chlorophyll-a), and the thickness of the ocean layer that is directly influenced by the atmosphere (mixed layer depth).

In total, the Pb concentration dataset includes 9920 observations, the $^{206}$Pb/$^{207}$Pb dataset 2014, and the $^{208}$Pb/$^{207}$Pb dataset 2010 (Fig. 1). In the WOA18 grid, 9920 observations cover ~0.3% of the total gridded ocean volume, while 2014 and 2010 cover

140 ~0.06%.

In the initial phase of model development, latitude and longitude were included as features and deconvoluted in a three-dimensional coordinate system to allow for continuity (Gregor et al., 2017; Huang et al., 2022). However, when including coordinates as features, the global maps of Pb concentration and isotope compositions showed unrealistic spatial artefacts (see SI Note 1). For this reason, coordinates were excluded as features for further model development and are not included in the

145 results presented here.

## 2.2 Model development and validation

The non-linear regression algorithm XGBoost (eXtreme Gradient Boosting; Chen & Guestrin, 2016) was chosen to develop three separate models for Pb concentration, $^{206}$Pb/$^{207}$Pb and $^{208}$Pb/$^{207}$Pb. Indeed, evidence shows that tree-based models consistently outperform deep learning models on tabular-style datasets (Grinsztajn et al., 2022). Moreover, in recent years,

XGBoost has proven to perform well on a variety of applications, spanning from finance to medicine, and including Earth sciences (Biass et al., 2022; Xie et al., 2024; Ye et al., 2023). XGBoost is an ensemble algorithm that consists of individual decision trees built in a sequential manner. In brief, each successive tree aims to reduce the errors perpetrated by its predecessor. The final prediction made by the model equals to the weighted sum of the predictions made by all trees in the ensemble. XGBoost was preferred to the Random Forest algorithm as the sequential building of trees in XGBoost allows the algorithm

to focus on poorly understood fields, and because Random Forest is more prone to overfitting than XGBoost. Moreover, the Random Forest algorithm was found to perform worse than XGBoost for the Pb concentration and $^{208}Pb/^{207}Pb$ models, and comparably for the $^{206}Pb/^{207}Pb$ model (SI Note 2).

In addition to performing well on a variety of applications, tree-based models also offer a higher level of interpretability compared to deep learning and linear models (Lundberg et al., 2020). Among the different approaches that have been used to

explain predictions of XGBoost models, Shapley Additive exPlanations (SHAP) have stood out in recent years as a unified approach to explain local predictions and gain a global understanding of a model's structure (Lundberg et al., 2020; Lundberg & Lee, 2017; Molnar, 2022).

The SHAP approach is based on game theory (i.e. features are considered as players that contribute to a prediction) and allows to interpret the importance of each feature on a given prediction by calculating the marginal contribution of each feature across

all possible permutations. For each sample, a positive (negative) SHAP value indicates that a specific feature contributes towards increasing (decreasing) the final predicted value. Additionally, features with a larger mean absolute SHAP value are considered to contribute more to a given prediction compared to features with a smaller mean absolute SHAP value.

In recent years, several studies in Earth sciences have used SHAP values for interpreting predictions made by tree-based and deep learning models, providing insightful findings on atmospheric pollution (Hou et al., 2022; Qin et al., 2022; Stirnberg et

al., 2021), ocean dynamics (Clare et al., 2022), and vegetation vulnerability (Biass et al., 2022). Here, we used the TreeExplainer method from the 'shap' library in Python (Lundberg et al., 2020; Lundberg & Lee, 2017) to compute SHAP values, and to evaluate the contribution of each feature on predicted Pb concentrations, $^{206}Pb/^{207}Pb$, and $^{208}Pb/^{207}Pb$. Interactions

between features, which are defined as the additional combined feature effects after accounting for the individual feature effects, were calculated using SHAP interaction values.

The first step of model development consisted of withholding a cruise transect in the Atlantic Ocean from the Pb concentration, $^{206}Pb/^{207}Pb$, and $^{208}Pb/^{207}Pb$ datasets. This group of withheld samples, identified as "geographic test set" (Fig. S1), included 216 samples for the Pb concentration model and 26 samples for the Pb isotope ratios models, and was used to mimic areas of the ocean where *in situ* samples have never been collected and assess the ability of the model to generalise to those. The remainder of each of the three datasets was split into a training test, consisting of 80% of the data, and a randomly selected test
set, consisting of 20% of the data.

To identify the best model architectures, hyperparameters were tuned using 5-fold cross-validation on the training set using the grid search method. Cross-validation is a technique that allows to evaluate the performance of a machine learning model during the hyperparameter tuning phase. It consists of splitting the training dataset into a specified number of subsets ($k$, in this case equal to 5), and using $k$-1 subsets for training and one subset for testing. The training and testing process is repeated
$k$ times (i.e. $k$-fold cross-validation), and the performance of each hyperparameter configuration is calculated by averaging the scores obtained for each fold. Hyperparameters are a set of parameters that control the learning process of a ML model and are tuned to obtain the most optimal model performance. The grid search method consists of specifying a set of possible values for each hyperparameter and subsequently training and testing the models built with each unique combination of hyperparameter values. The hyperparameters tuned for our Pb concentration, $^{206}Pb/^{207}Pb$, and $^{208}Pb/^{207}Pb$ include: the number
of decision trees built and boosted ('*n_estimators*'), the rate at which the model learns information ('*learning_rate*'), the maximum number of split nodes in a tree ('*max_depth*'), the minimum number of samples a node must represent in order to be split further ('*min_child_weight*'), the percentage of features used to construct each tree ('*colsample_bytree*'), and the L1 and L2 regularization terms ('*reg_alpha*' and '*reg_lambda*', respectively). The two regularization terms, L1 and L2, were only tuned for the Pb isotope ratio models to reduce the computational costs of hyperparameter tuning for the Pb concentration
model, which was built using a much larger dataset than the $^{206}Pb/^{207}Pb$ and $^{208}Pb/^{207}Pb$ models. As the target variables of the three models (Pb concentrations, $^{206}Pb/^{207}Pb$, and $^{208}Pb/^{207}Pb$) were not uniformly distributed, a least-square loss function was

chosen as the preferred method for model optimization, as squaring the error penalises the model more when the size of the error increases.

Different performance metrics were used to assess model performance during cross validation and testing on the geographic
and random test sets. These include root mean square error (RMSE), mean average percentage error (MAPE), and the coefficient of determination ($R^2$), and are calculated as follows:

$$RMSE = \sqrt{\frac{1}{n}\sum_{i=1}^{n}(y_o - y_p)^2}$$

$$MAPE = 100\frac{1}{n}\sum_{i=1}^{n}\left|\frac{y_o - y_p}{y_o}\right|$$

$$R^2 = 1 - \frac{\sum_{i=1}^{n}(y_o - y_p)^2}{\sum_{i=1}^{n}(y_o - \bar{y})^2}$$

where $n$ is the number of samples, $y_o$ and $y_p$ the observed and predicted value, respectively, and $\bar{y}$ is the mean of the observed values.

Root mean square error was chosen as the primary evaluation metric for the two isotope ratio models. Indeed, RMSE is reported in the same unit as the target variables (in this case unitless) and makes it more intuitive to interpret the results. On the other hand, MAPE was chosen as the preferred metric on which to evaluate the performance of Pb concentration models. This was
done as Pb concentrations in the dataset vary between 0.10 and 95.00 pmol/kg, with an average value of $17.33 \pm 13.05$ (1SD, n = 9920). While MAPE might be harder to interpret, as it is reported as a percentage rather than an absolute value with the same unit of measurement as the target variable, minimising the percentage error ensures that basins and locations where Pb concentrations are low are not overlooked by the model.



# 3 Results and discussion

## 3.1 Model development and performance

During hyperparameter tuning, we assessed model configurations by changing hyperparameter grid spacing and increasing granularity to get as close as possible to the best combination achievable. In the final phase of the tuning, 1280 different configurations were assessed for Pb concentrations, and 11520 both for $^{206}Pb/^{207}Pb$ and $^{208}Pb/^{207}Pb$. The hyperparameter space explored and the best hyperparameters identified for each of the three models are reported in Table 2. The variations in MAPE and RMSE values for different combinations of hyperparameters are visible in Fig. S2.

The mean MAPE values of the Pb concentration models calculated on the cross-validation, random test set and geographic test set were $22.7 \pm 2.2$ %, $20.7 \pm 1.5$ %, and $19.5 \pm 4.1$ % (n = 1280, 2SD), respectively. The ten best hyperparameter configurations for the Pb concentration model shared the same learning rate (0.01) and number of columns subsampled by each tree (0.7). The maximum difference in MAPE values between these ten models was 0.05 % on the cross-validation set, 0.17 % on the random test set, and 0.82 % on the geographic test set.

**Table 2. Hyperparameter space explored for the XGBoost regression models. Bold values identify the combination of hyperparameters that returned the best model performance.**

| Hyperparameter | Pb concentration | | | | | $^{206}Pb/^{207}Pb$ | | | | | $^{208}Pb/^{207}Pb$ | | | | |
|---|---|---|---|---|---|---|---|---|---|---|---|---|---|---|---|
| learning_rate | **0.01** | 0.05 | 0.1 | 0.3 | | **0.01** | 0.05 | 0.1 | | | **0.01** | 0.05 | 0.1 | | |
| n_estimators | 1200 | 1300 | 1400 | **1500** | 1600 | 1100 | 1200 | 1300 | **1400** | 1500 | 700 | 800 | **900** | 1000 | 1100 |
| max_depth | 12 | **14** | 16 | 18 | | 4 | 6 | **8** | 10 | | 16 | 18 | **20** | 22 | |
| colsample_bytree | 0.5 | **0.7** | 0.9 | 1 | | 0.5 | **0.7** | 0.9 | 1 | | **0.5** | 0.7 | 0.9 | 1 | |
| min_child_weight | 6 | **8** | 10 | 12 | | 6 | 8 | **10** | 12 | | 12 | 14 | **16** | 18 | |
| reg_alpha | | | | | | **0** | 0.1 | 1 | | | **0** | 0.1 | 1 | | |
| reg_lambda | | | | | | 0 | **0.1** | 1 | 10 | | 6 | 8 | **10** | 12 | |

The $^{206}Pb/^{207}Pb$ models had mean RMSE values of $0.008 \pm 0.004$ (cross-validation), $0.007 \pm 0.003$ (random test set), and $0.006 \pm 0.002$ (geographic test set; n = 11520, 2SD). The 20 best performing hyperparameter configurations shared the same learning rate (0.01), L1 regularization (0) and number of columns subsampled by each tree (0.7). These models have a maximum difference in RMSE values of 0.000024 on the cross-validation set, 0.000194 on the random test set, and 0.000298 on the geographic test set. Lastly, the $^{208}Pb/^{207}Pb$ models had mean RMSE values of $0.007 \pm 0.003$ (cross-validation), $0.007 \pm 0.003$ (random test set), and $0.008 \pm 0.003$ (geographic test set; n = 11520, 2SD). The best five hyperparameter configurations have



the same learning rate (0.01), number of columns subsampled by each tree (0.5), and L1 regularization (0). The maximum

difference in RMSE values between these models was 0.000003 on the cross-validation set, 0.000046 on the random test set,

and 0.000069 on the geographic test set.

Overall, given the very comparable performances of the best hyperparameter configurations on cross-validation, random test

set and geographic test set for each of the three models described above, we built the final Pb concentration, $^{206}$Pb/$^{207}$Pb, and

$^{208}$Pb/$^{207}$Pb models using the hyperparameter configuration that performed best on cross-validation.

**Table 3. Performance of the best Pb concentration, $^{206}$Pb/$^{207}$Pb, and $^{208}$Pb/$^{207}$Pb models.**

| Target variable | Random test set | | | | Geographic test set (Atlantic Ocean) | | | |
|---|---|---|---|---|---|---|---|---|
| | $n$ | $R^2$ | RMSE | MAPE [%] | $n$ | $R^2$ | RMSE | MAPE [%] |
| [Pb] | 1941 | 0.87 | 4.84 pmol/kg | **20.2** | 216 | 0.82 | 5.06 pmol/kg | **18.3** |
| $^{206}$Pb/$^{207}$Pb | 398 | 0.80 | **0.006** | 0.3 | 26 | 0.77 | **0.005** | 0.3 |
| $^{208}$Pb/$^{207}$Pb | 397 | 0.71 | **0.006** | 0.1 | 26 | 0.52 | **0.006** | 0.2 |

The predictive performance of the final models was assessed on both the random and geographic test sets and is reported in

Table 3. For all three models, the agreement between observed and predicted values ($R^2$ score) is higher for the random test

set than for the geographic test set. However, MAPE and RMSE are not always better for the random test set than for the

geographic one, indicating that the model is able to generalise well without drastically reducing its performance. Indeed, the

Pb concentration model has a lower MAPE for the geographic test set (18.3 %) than for the random one (20.2 %), and the

$^{206}$Pb/$^{207}$Pb model has a lower RMSE for the geographic (0.005) than the random test set (0.006).

Model performance on the random test set was also assessed by splitting the data between ocean basins to identify which areas

are better reproduced by the models (Fig. 2). The Pb concentration model achieves the highest $R^2$ on the Pacific Ocean (0.93)

and the lowest MAPE on the Atlantic Ocean (14 %). The model fitting of Indian Ocean observations is overall very good ($R^2$

= 0.84), however the MAPE is relatively high at 33 %, mostly due to a poor performance on data between 25 and 65 pmol/kg.

Lastly, the two high-latitude basins have the worst $R^2$ (Arctic Ocean: 0.49; Southern Ocean: 0.58) and high MAPE values

(Arctic Ocean: 37 %; Southern Ocean: 29 %), due to the presence of outliers as well as a high density of low concentration



values (i.e. MAPE is higher for a given difference between modelled and actual value when observation values are closer to

zero).



**Figure 2. Basin-wise model performances assessed on all samples from each basin in the random test set. Top row: Pb**
**concentration; Middle row: $^{206}Pb/^{207}Pb$; Bottom row: $^{208}Pb/^{207}Pb$.**

The Pb isotope ratio models perform the worst on the Arctic Ocean ($^{206}Pb/^{207}Pb$: $R^2 = 0.52$, RMSE = 0.010; $^{208}Pb/^{207}Pb$: $R^2 =$

0.55, RMSE = 0.011), and show discrepant performances on the Atlantic, Pacific, Indian and Southern Ocean. The $^{206}Pb/^{207}Pb$

model performs best on the two basins with the lowest number of samples in the random test, namely the Southern ($R^2 = 0.93$,

RMSE = 0.004) and Indian Ocean ($R^2 = 0.87$, RMSE = 0.004). Samples from the Pacific Ocean are well reproduced ($R^2 =$



0.80, RMSE = 0.004), while the performance on the Atlantic Ocean is poorer ($R^2$ = 0.59, RMSE = 0.006). Lastly, for the

$^{208}$Pb/$^{207}$Pb model, the Indian Ocean is the basin with the best performance ($R^2$ = 0.89, RMSE = 0.003), followed by the Atlantic

($R^2$ = 0.74, RMSE = 0.004) and the Pacific ($R^2$ = 0.57, RMSE 0.005).

### 3.2 Model explanation

SHAP values were calculated for all samples in the Pb concentration, $^{206}$Pb/$^{207}$Pb, and $^{208}$Pb/$^{207}$Pb datasets to maximise the

interpretability of the models. In the next subsections, the four most important features for each model, as well as their

interaction terms, are explained.

**Table 4. SHAP importance values for all features included in the Pb concentration, $^{206}$Pb/$^{207}$Pb, and $^{208}$Pb/$^{207}$Pb models.**

| Feature Name | *Pb concentration* | | *$^{206}$Pb/$^{207}$Pb* | | *$^{208}$Pb/$^{207}$Pb* | |
|---|---|---|---|---|---|---|
| | Importance | Rank | Importance | Rank | Importance | Rank |
| Apparent oxygen utilization [μmol/kg] | 0.22 | 14 | 0.00033 | 14 | 0.00025 | 14 |
| Black Carbon AOD | **2.01** | **3** | 0.00141 | 6 | **0.00185** | **1** |
| Bottom distance [m] | 0.49 | 11 | 0.00074 | 10 | 0.00091 | 5 |
| Chlorophyll-a [mg/m$^3$] | 0.63 | 9 | 0.00070 | 11 | **0.00097** | **4** |
| Density [kg/m$^3$] | 0.76 | 5 | 0.00144 | 5 | 0.00064 | 10 |
| Depth [m] | 0.68 | 8 | 0.00086 | 7 | 0.00073 | 8 |
| Dust AOD | **2.84** | **2** | **0.00154** | **4** | **0.00114** | **2** |
| Mixed layer depth [m] | 0.71 | 7 | 0.00074 | 9 | 0.00079 | 6 |
| Nitrate [μmol/kg] | 0.41 | 13 | 0.00034 | 13 | 0.00048 | 13 |
| Oxygen [μmol/kg] | 0.46 | 12 | **0.00191** | **1** | 0.00078 | 7 |
| Phosphate [μmol/kg] | 0.56 | 10 | 0.00064 | 12 | 0.00049 | 12 |
| Salinity | **0.99** | **4** | **0.00170** | **2** | 0.00057 | 11 |
| Silicate [μmol/kg] | 0.74 | 6 | 0.00083 | 8 | 0.00069 | 9 |
| Temperature [°C] | **3.19** | **1** | **0.00155** | **3** | **0.00102** | **3** |

### 3.2.1. Pb concentrations

Temperature, dust AOD, black carbon AOD, and salinity are the most important features for the Pb concentration model (Table

4, Fig. 3). For temperature values above 5.8 °C, SHAP values are consistently positive, with warmer waters having a larger

impact on predicted Pb concentrations (Fig. 3A). Black carbon AOD has the largest interaction with temperature, showing

positive SHAP values in areas where both features are high. Taken together, these results suggest that surface and intermediate

waters at low and mid latitudes (where black carbon AOD is highest) are associated with higher predictions of Pb

concentrations. An exception to this trend is observed for samples with low temperatures and extremely low black carbon



AOD, which have unexpectedly positive SHAP values for temperature (Fig. 3A). Analogously, SHAP values for black carbon

AOD are positive for the lowest black carbon AOD values (< 0.001; Fig. 3C). The positive impact of extremely low black

carbon concentrations in the atmosphere cannot be explained by a physico-chemical process, as one would expect lower values

of black carbon AOD to lead to lower Pb concentrations in the ocean. However, the positive SHAP values for temperature and

black carbon AOD can be explained to the presence of samples with high Pb concentrations in the Southern Ocean (Fig. 1A),

where black carbon AOD is lowest, and seawater temperatures are below 5.0 °C (Fig. S3 and S4). Further evidence supporting

the positive impact of the lowest black carbon AOD values on Pb concentrations in the Southern Ocean can be found in the

interaction between black carbon AOD and salinity observed for salinity SHAP values (Fig. 3D). Indeed, this interaction

indicates positive SHAP values for extremely low black carbon AOD and salinities around 34.0 ± 0.2 (Fig. S5), which are

characteristics of the higher latitudes of the Southern Hemisphere.

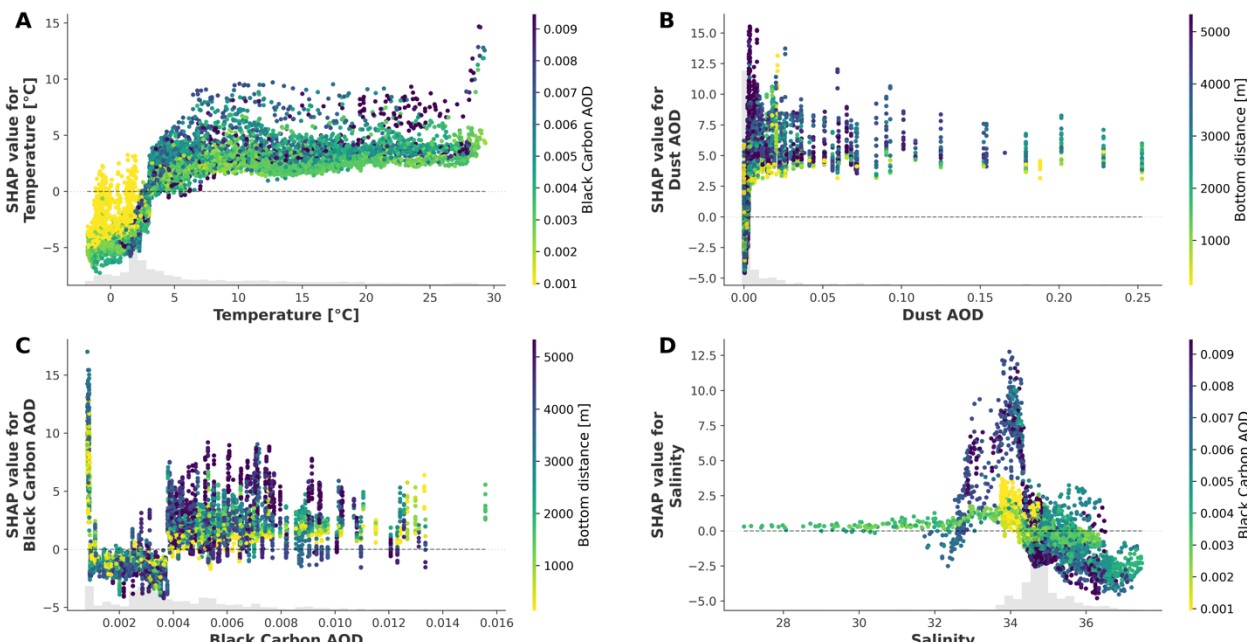

**Figure 3. SHAP values for temperature (A), dust AOD (B), black carbon AOD (C), and salinity (D) for the Pb concentration model. The colour of the dots represents the value of the feature that has the highest interaction (reported on the right-hand-side of each plot). Y-axis values differ among the panels.**



SHAP values are consistently positive for dust AOD values above 0.003 (Fig. 3B), and they show an interesting interaction

with distance from bottom depth, which is especially visible for larger values of dust AOD (> 0.05). Indeed, as all grid-cells

at the same location were assigned the same dust AOD value throughout the water column, the interaction between dust AOD

and distance from bottom depth shows that samples more distant from the seafloor (i.e. closer to the surface) have higher

SHAP values than their deeper counterparts (Fig. 3B). This finding agrees with observations of higher Pb concentrations at

the surface ocean due to atmospheric deposition (Duce et al., 1991; Patterson & Settle, 1987), therefore suggesting that the

model has learned from the data the key process of atmospheric Pb sourcing to the ocean.

### 3.2.2. Pb isotope compositions

Seawater oxygen concentration, salinity, temperature, and dust AOD are the four most important features for the $^{206}Pb/^{207}Pb$

model (Table 4; Fig. 4). Oxygen concentration values below 220 µmol/kg have SHAP values between -0.004 and 0, and a

sharp transition to positive SHAP values can be observed at oxygen concentrations of ~240 µmol/kg (Fig. 4A). The strong

interaction between oxygen and nitrate concentrations suggests that areas of high dissolved oxygen and low nitrate

concentrations, such as the Arctic and the North Atlantic Ocean (Fig. S6 and S7), have a large positive impact on predicted

$^{206}Pb/^{207}Pb$.

SHAP values for salinity show an increasing trend as the latter varies between 32.5 and 37.0 (Fig. 4B). This is especially

visible for surface and intermediate waters with densities ($\sigma_0$) between 25 and 35 km/m$^3$, indicating that the model learned

well from observations, which show higher Pb concentrations at shallower depths and a general decrease in Pb concentration

values from tropical and subtropical to polar latitudes (Fig. 1A). SHAP values for temperature show a decreasing trend as

waters get warmer (Fig. 4C). Interestingly, and expectedly from a theoretical perspective, this trend is reversed for density,

which is the fifth most important feature for the model and shows increasing SHAP values with as waters become denser (Fig.

S8). Both temperature and density also show a clear interaction with salinity, which can be explained by the fact that

temperature and salinity are the two conservative parameters that contribute to seawater density. Overall, the SHAP values of

and interactions between these three variables agree with the general view of Pb isotope ratios as tracers of water mass

movements and ocean ventilation (Bridgestock et al., 2018; Frank, 2002; Henderson & Maier-Reimer, 2002).

Dust AOD shows a more variable distribution of SHAP values, with strongly positive SHAP values (up to 0.009) for dust

AODs lower than 0.004, followed by a rapid decrease to strongly negative SHAP values (up to -0.007) for dust AODs between

0.004 and 0.007 (Fig. 4D). As dust AOD increases, SHAP values show a flat trend, followed by a slight decrease between dust

AODs of 0.05 and 0.12, and an increase afterwards (Fig. 4B). The interaction between dust AOD and silicate highlights that

for dust AOD values above 0.007, SHAP values tend to be higher where silicate concentrations are lower, such as in the surface

layer (besides in the Southern Ocean and northern North Pacific), and the intermediate and deep Arctic and North Atlantic

Ocean (Fig. S9 and S10).

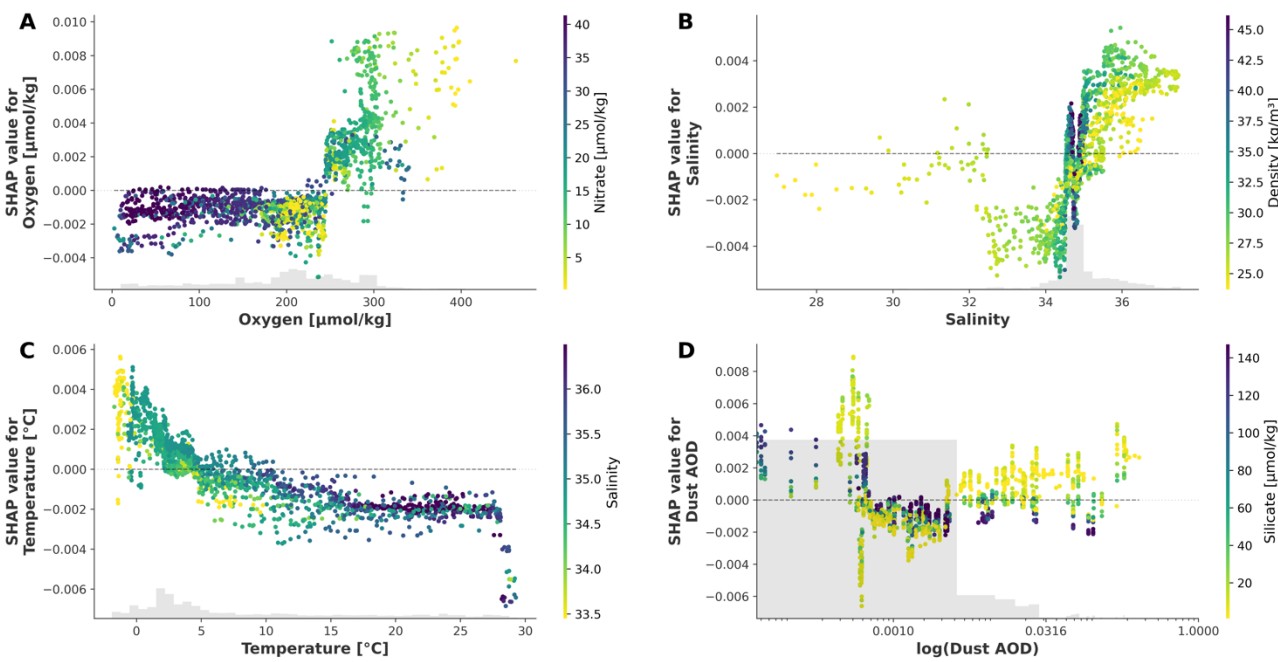

**Figure 4. SHAP values for oxygen (A), salinity (B), temperature (C), and dust AOD (D – on a logarithmic scale) for the**
**$^{206}$Pb/$^{207}$Pb model. The colour of the dots represents the value of the feature that has the highest interaction (reported**
**on the right-hand-side of each plot). Y-axis values differ among the panels.**

Perhaps surprisingly, given the generally observed correlation between the $^{206}$Pb/$^{207}$Pb and $^{208}$Pb/$^{207}$Pb isotope ratios (Boyle et

al., 2012; GEOTRACES Intermediate Data Product Group, 2023), the most important features for $^{208}$Pb/$^{207}$Pb (Fig. 5) do not

match closely those of $^{206}$Pb/$^{207}$Pb (Fig. 4). In fact, the four most important features for the $^{208}$Pb/$^{207}$Pb model include black

carbon AOD, dust AOD, seawater temperature, and surface chlorophyll-a concentration (Table 4; Fig. 5). Black carbon AOD

has mostly positive SHAP values for feature values above 0.0045 and negative otherwise (Fig. 5A). It has the strongest

interaction with surface chlorophyll-a, and vice versa. Moreover, SHAP values for chlorophyll-a are mostly negative for the

lowest chlorophyll-a concentrations, and range between 0.003 and -0.002 for chlorophyll-a concentrations above 0.13 mg/m$^3$

(Fig. 5D). The interaction between these two features, however, does not show a clear pattern and cannot be interpreted

intuitively, as both high and low chlorophyll-a concentrations have high and low SHAP values for black carbon AOD, and the

other way round (Fig. 5A and 5D). SHAP values of dust AOD (Fig. 5B) show a similar pattern to the $^{206}$Pb/$^{207}$Pb model, with

lower dust AOD values having both the highest and lowest SHAP values. The interaction between dust AOD and black carbon

AOD suggests that when both features are low, dust AOD has a strong positive impact on predicted $^{208}$Pb/$^{207}$Pb. Contrarily,

when dust AOD is low and black carbon AOD higher than 0.004, dust AOD contributes to reducing predicted $^{208}$Pb/$^{207}$Pb

values (Fig. 5B). Lastly, SHAP values for temperature also show a similar decreasing trend to those of the $^{206}$Pb/$^{207}$Pb model

and have a strong interaction with dissolved nitrate concentrations (Fig. 5C). This interaction indicates that warmer waters

with low nitrate content, such as those in tropical and subtropical areas (Fig. S4 and S7), are associated with a reduction in

predicted $^{208}$Pb/$^{207}$Pb values.

### 3.2.3. SHAP model explanations and current understanding of marine Pb cycling

SHAP explanations for the Pb concentration, $^{206}$Pb/$^{207}$Pb, and $^{208}$Pb/$^{207}$Pb models align well the current understanding of marine

biogeochemical cycling of Pb, with respect to the oceanographic predictors identified as most important features for the three

models. This indicates that the models are able to identify the key oceanographic patterns and processes that drive the

distribution of Pb and its isotopes on a global scale. However, some regional explanations of dust and black carbon AODs for

the Pb concentration and isotope composition models appear to contradict the theoretical intuition that dust and black carbon

are sources of Pb to the ocean, with Pb isotope compositions broadly reflecting those of natural and anthropogenic land sources,

respectively. In particular, the high SHAP values obtained for extremely low black carbon AODs in the Pb concentration

model disagree with the hypothesis that low black carbon concentrations in the atmosphere should result in lower Pb

concentrations in the ocean. Analogously, the highly positive SHAP values obtained for extremely low values of dust AOD

for the $^{206}$Pb/$^{207}$Pb and $^{208}$Pb/$^{207}$Pb models, and the consistently positive SHAP values obtained for relatively high black carbon

AODs for the [208]Pb/[207]Pb model do not match the intuition that dust should contribute to shifting the Pb isotope composition

of seawater toward more natural values and black carbon toward more anthropogenic ones.

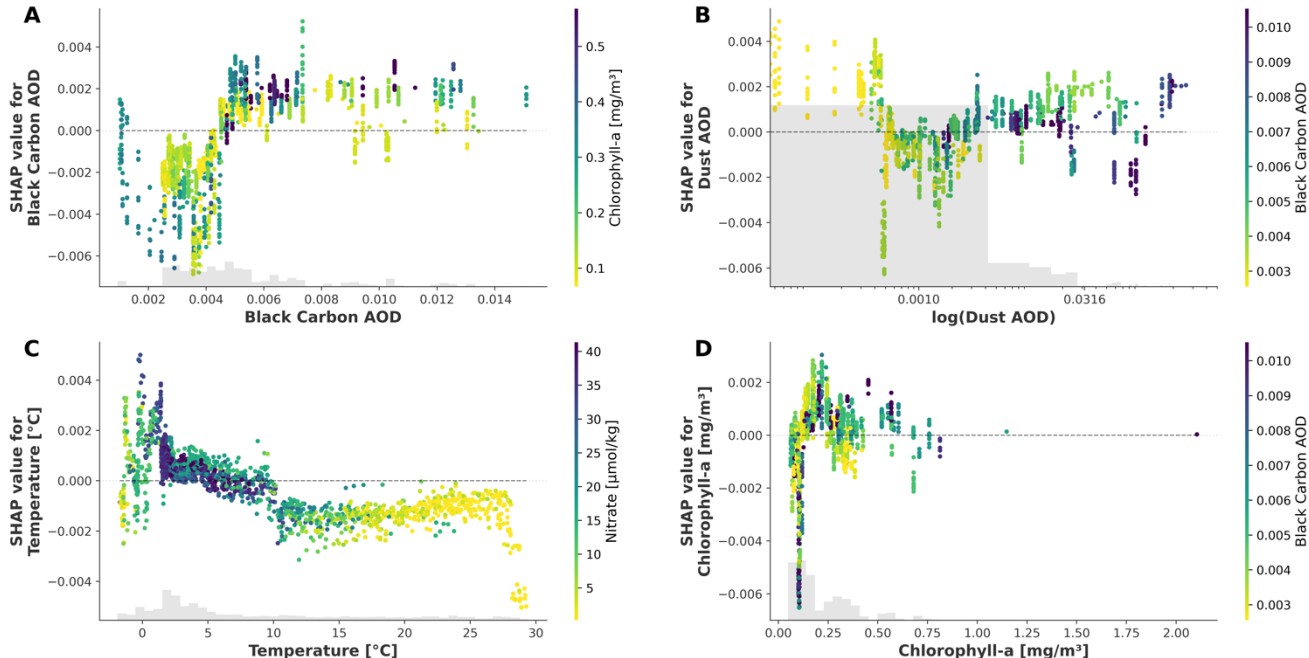

**Figure 5. SHAP values for black carbon AOD (A), dust AOD (B – on a logarithmic scale), temperature (C), and surface chlorophyll-a (D) for the [208]Pb/[207]Pb model. The colour of the dots represents the value of the feature that has the highest interaction (reported on the right-hand-side of each plot). Y-axis values differ among the panels.**

Possible explanations for the partial mismatch between expected and obtained local SHAP values are provided in the following,

but these will need validation through new campaigns at sea and further studies of the Pb isotope compositions of dust and

black carbon sources to the atmosphere. A first explanation is that dust and black carbon are not the only sources of Pb to the

global ocean and this might have important implications on a regional scale. An example is the Southern Ocean, where ice

melting and runoff could contribute significantly to the marine Pb budget of the area. Another possibility is that global

climatologies of dust AOD and black carbon AOD do not capture the regional and temporal variability of the Pb isotope

compositions of natural and anthropogenic Pb sources to the environment. Moreover, black carbon is not only emitted due to

fossil fuel consumption, but also from forest fires and coal combustion, which might have substantially different Pb isotope

signatures compared to other industrial and urban sources. Therefore, while the global climatologies of dust AOD and black



carbon AOD used as model predictors are currently the best available proxies for atmospheric sources of Pb, they might not

fully encompass the complexity of Pb sources to the environment, resulting in locally counterintuitive SHAP explanations for

the three models.

### 3.3. Global maps of Pb concentrations and isotope compositions in seawater

Global reconstructions of Pb concentrations, $^{206}$Pb/$^{207}$Pb, and $^{208}$Pb/$^{207}$Pb were made for each grid cell in the WOA18 using the

same structure and ancillary features used for model development (Sect. 2.1).

Figure 6 shows the global distribution of Pb concentration, $^{206}$Pb/$^{207}$Pb, and $^{208}$Pb/$^{207}$Pb at four depth levels (10 m, 1000 m,

2500 m, and 4000 m), arbitrarily chosen to represent surface, intermediate, deep, and bottom waters, respectively. These maps

provide the first global-scale estimates of Pb concentrations and isotopes and can be used to further understand the dynamics

and mechanisms that govern the large-scale distribution of Pb.

### 3.3.1. Pb concentrations

Lead concentrations vary between 2.05 and 77.65 pmol/kg. Generally, concentrations are higher at the surface and show a

decreasing trend with depth (Fig. 6A). The only exception to this trend is the North Atlantic Ocean, where mapped Pb

concentrations at 1000 m and 2500 m are higher than at 10 m. Both the general trend and the exception of the North Atlantic

are in line with the expected distribution of Pb concentrations. Indeed, the majority of Pb is sourced to the ocean via

atmospheric wet deposition and then sinks towards the bottom aided by particulate matter (Duce et al., 1991; Patterson &

Settle, 1987). Therefore, high Pb concentrations are expected for the surface ocean, with values decreasing with depth.

However, extensive historical pollution from leaded gasoline in North America and Europe (Kelly et al., 2009) led to high Pb

concentrations in North Atlantic Ocean surface waters between the 1950s and 1980s. Lead concentrations eventually decreased

due to environmental policies that phased out and banned leaded gasoline, and Pb pollution in surface waters has decreased

steadily since 1975 (Boyle et al., 2014; Bridgestock et al., 2016; Kelly et al., 2009). Therefore, the relatively high Pb

concentrations at intermediate- and deep-water depths reflects the sinking of elemental Pb pollution from earlier decades (Fig.

6A), in agreement with observations (Fig. 1A; Noble et al., 2015).



**Figure 6. Global maps of reconstructed Pb concentrations (A) and 206Pb/207Pb (B), and 208Pb/207Pb (C). The four panels represent different depth levels (10 m, 1000 m, 2500 m, 4000 m), with white patches corresponding to seafloor. The filled circles with white edges represent true data values for comparison with the modelled values in the background.**

Mapped Pb concentrations are highest in surface Indian Ocean (average [Pb] = 32.0 ± 14.2 pmol/kg, 1SD) and in the North

Pacific (33.6 ± 11.3 pmol/kg, 1SD), in line with recent observations from the two basins (Chen et al., 2023a; Echegoyen et al.,

2014; Lanning et al., 2023). The North Pacific also shows high mapped Pb concentrations at 1000 m depth (Fig. 6A), which

can be reconciled with pollution from previous decades (Lanning et al., 2023; Wu et al., 2010), like in the North Atlantic.

Relatively high Pb concentrations are furthermore found at the surface level near South and Central America, Western Africa,

and Southeast Asia. Except for samples collected close to the coast in the South Atlantic Ocean, these areas have never been

sampled before and the high concentrations predicted by the model cannot be directly verified. However, the distribution of

predictor features shows that these are all areas with high seawater temperatures, and relatively high black carbon and dust

AODs (Fig. S4, S3 and S9). Therefore, following the model's interpretation with SHAP values (Sect. 3.2.1), these features

must be strongly contributing to the high predictions made by the model.

Lastly, the polar regions are characterised by the lowest Pb concentrations throughout the water column, with average Pb

concentrations of 9.0 ± 3.1 pmol/kg (1SD) for the Arctic Ocean and 11.9 ± 3.6 pmol/kg (1SD) for the Southern Ocean.

Additionally, both polar oceans have the least variable Pb concentrations with depth (Fig. S11), which denotes them as the

areas of the global ocean that are least affected by anthropogenic pollution.

### 3.3.2. Pb isotope compositions

Global maps of $^{206}$Pb/$^{207}$Pb and $^{208}$Pb/$^{207}$Pb ratios show similar patterns at the four different depth levels analysed (Fig 6B, C).

Globally, $^{206}$Pb/$^{207}$Pb ratios vary between 1.144 and 1.200, and $^{208}$Pb/$^{207}$Pb between 2.420 and 2.479. Generally, lower

$^{206}$Pb/$^{207}$Pb and $^{208}$Pb/$^{207}$Pb values are associated with Pb sourced from anthropogenic activities, while higher values reflect

natural sources of Pb (Bollhöfer & Rosman, 2000, 2001). However, the exact isotope composition of natural and anthropogenic

Pb varies between sources and basins, and there is no single reference value that is valid on a global scale due to the short

residence time of Pb in seawater and the heterogeneity of global Pb sources. Recent work has, furthermore, shown that

reversible scavenging can affect and modify the advected Pb isotope signature in areas of high suspended particulate matter

(Lanning et al., 2023; Olivelli et al., 2024b). However, historically Pb isotopes have been regarded as tracers of water mass

movements and ventilation (e.g. Véron et al., 1998, 1999). We therefore additionally assessed the distribution of $^{206}$Pb/$^{207}$Pb and $^{208}$Pb/$^{207}$Pb across different ranges of potential density anomaly ($\sigma_\theta$; Fig. 6 and Fig. S12).

The median $^{206}$Pb/$^{207}$Pb value is highest for the Arctic (1.186) and lowest for the Indian Ocean (1.165; Fig 7). Median values for the North and South Pacific, South Atlantic and Southern Ocean are very similar, varying between 1.168 and 1.171, while the North Atlantic Ocean has a higher median $^{206}$Pb/$^{207}$Pb of 1.180. Additionally, the Arctic Ocean shows the smallest difference between the first and third quartile, indicating that Pb isotope compositions are rather constant throughout the geographical domain and the water column, while the Southern Ocean has the most variable distribution (Fig. 6).

In the surface layer (10 m), the lowest $^{206}$Pb/$^{207}$Pb and $^{208}$Pb/$^{207}$Pb values are observed in the northern Indian Ocean, while the Arctic Ocean has the highest $^{206}$Pb/$^{207}$Pb values and coastal areas around North America, the Arctic Ocean and the eastern Pacific Ocean have the highest $^{208}$Pb/$^{207}$Pb values (Fig. 6B, C). Additionally, the distribution of both isotope ratios shows the presence of low values ($^{206}$Pb/$^{207}$Pb: 1.148 – 1.160; $^{208}$Pb/$^{207}$Pb: 2.420 – 2.437) at 1000 m depth, with a core at the location of the Subantarctic Front (~48 °S), expanding horizontally to all basins in the Southern Hemisphere and vertically to the deep

layer (2500 m). Low $^{206}$Pb/$^{207}$Pb and $^{208}$Pb/$^{207}$Pb values at intermediate depths have been observed in the Indian (Lee et al., 2015), South Atlantic (Olivelli et al., 2024b), and Southern Ocean off the coast of Tasmania (GEOTRACES cruise GS01; unpublished data from the MAGIC group at Imperial College London). Historically, anthropogenic emissions from Australia, New Zealand, Chile and South Africa record the lowest Pb isotope ratios and are mostly associated with the use of Broken Hill-type leaded gasoline (with $^{206}$Pb/$^{207}$Pb and $^{208}$Pb/$^{207}$Pb as low as 1.060 and 2.328, respectively; Bollhöfer & Rosman,

2000). Our maps, therefore, suggest that Antarctic Intermediate Water (AAIW), which sinks to depth at latitudes between 50 and 60 °S across the Southern Ocean, in the eastern South Pacific and near the Drake Passage, has the most anthropogenic signature of all non-surface water masses considered (Fig. 6B, C). Further evidence for the highly anthropogenic signature of AAIW can be found by analysing the distribution of Pb isotope compositions across density layers. Indeed, AAIW, which has a potential density anomaly between 27.0 – 27.4 kg/m$^3$, corresponds to the layer with the lowest median $^{206}$Pb/$^{207}$Pb in the

South Pacific, South Atlantic, and Indian Ocean (Fig. 7). As leaded gasoline was phased out in the abovementioned countries, and their neighbours, before the time of sampling on GEOTRACES cruises, the widespread anthropogenic signature observed



**Figure 7. Violin plots of $^{206}$Pb/$^{207}$Pb distributions in the different ocean basin (top left panel) and within each basin at different potential density ranges. The white dashed line in each violin represents the median value, while the dotted lines represent the lower and upper quartiles (Q1 and Q3, respectively).**

at 1000 m in the Southern Hemisphere either reflects persisting pollution from previous decades that was deposited in the Southern Ocean and is transported northward or the remobilisation of previously deposited Pb on land and subsequent atmospheric deposition in the formation areas of AAIW.

In the North Pacific, North Atlantic and Southern Ocean, the median $^{206}Pb/^{207}Pb$ and $^{208}Pb/^{207}Pb$ values increase with depth and potential density ranges (Fig 6B, C and Fig. 7). Contrarily, in the South Atlantic, Indian, and, to a lesser extent, the South

Pacific Ocean, a decrease in $^{206}Pb/^{207}Pb$ values can be observed between surface and intermediate water masses, generally followed by an increase as water masses become denser (Fig. 7). The mapped distributions of Pb isotope compositions, taken together with that of Pb concentrations, therefore indicate that the deepest and densest water masses are least affected by anthropogenic pollution on a global scale, as they have the lowest Pb concentrations and highest $^{206}Pb/^{207}Pb$ and $^{208}Pb/^{207}Pb$ ratios. On the other hand, surface and intermediate waters are most affected by pollution, which is discernible (i) in the Northern

Hemisphere from Pb concentration data and to some extent from Pb isotope compositions, although natural and anthropogenic sources affecting those basins have a relatively small difference in Pb isotope fingerprints, and (ii) in the Southern Hemisphere, and the northern Indian Ocean, from the low ratios of $^{206}Pb/^{207}Pb$ and $^{208}Pb/^{207}Pb$.

### 3.4. Uncertainty quantification

Three different sources of uncertainty were identified for our model outputs and are described, and where possible quantified,

in this section. The first source of uncertainty relates to the observed values of Pb concentrations and isotope compositions, the second to the ancillary variables used as features in the models, and the third to the random split of the data into a training and a test set.

All observations of Pb concentrations, $^{206}Pb/^{207}Pb$, and $^{208}Pb/^{207}Pb$ ratios are characterised by a fundamental level of uncertainty associated with measuring Pb and its isotopes in seawater. While the GEOTRACES programme ensures data quality of the

highest standard through a thorough process of intercalibration, it cannot eliminate uncertainty that is intrinsic to sample processing and analysis. In the case of Pb concentrations, individual sample uncertainties, arising from replicate measurements generally range between 0.1% and 80%, and on average are around 10 to 25 % (relative 1SD; e.g. Boyle et al., 2020; Echegoyen et al., 2014). Inter-laboratory uncertainties arising from the use of different sample processing methodologies and different

instrumentation and assessed on intercalibration samples, however, typically show a better reproducibility. In fact, the relative

standard deviation of the North Atlantic GEOTRACES and SAFe reference samples is between 3.5% and 9.4% (1SD; GEOTRACES, 2009, 2013). For Pb isotope compositions, individual sample uncertainties, assessed either as replicate measurements of the sample itself or extrapolated from replicates measurements of NIST 981 standard reference material is generally around 0.2 – 0.5‰ for both $^{206}Pb/^{207}Pb$ and $^{208}Pb/^{207}Pb$ (e.g. Boyle et al., 2020; Bridgestock et al., 2016). Inter-laboratory calibration exercises of Pb isotope composition measurements of seawater have returned uncertainties ranging

between 0.1‰ and 3.5‰ for both isotope ratios considered (Boyle et al., 2012). So, if we consider the MAPE as a reflection of the uncertainty associated with the models' predictions, predicted values of Pb concentration have an uncertainty (20.2% random test set, 18.3% geographic test set; Table 3) that falls within the average values for replicate measurements $(10 - 25\%)$. Similarly, the uncertainty associated with predicted $^{206}Pb/^{207}Pb$ and $^{208}Pb/^{207}Pb$ values (MAPE: $1.0 - 3.0‰$; Table 3) falls within the range observed for inter-laboratory calibration.

Pooling together dissolved and total dissolvable Pb concentration and isotope composition data creates an additional layer of uncertainty. Our choice is backed by the evidence that in the open ocean, dissolved and total dissolvable Pb data are comparable (Bridgestock et al., 2018; Olivelli et al., 2023, 2024b; Schlosser et al., 2019). However, this might not be the case in estuaries and coastal areas, where boundary exchange and isotopic re-equilibration are believed to play an important role in determining the observed Pb concentrations and isotope compositions (Chen et al., 2016, 2023b). Therefore, new observations, particularly

from coastal areas and other sparsely sampled locations, would provide valuable data for the validation of our modelling results.

The uncertainty associated with the models' features cannot be quantified directly, and contributes to the models' MAPE values, but is nevertheless expected to be small and not create systematic bias. Indeed, the data products used in this study, namely WOA18 for oceanographic variables, CMEMS's Global Ocean Biogeochemistry Hindcast product for chlorophyll-a,

and CAMS's ECMWF Atmospheric Composition Reanalysis 4 for black carbon AOD and dust AOD, are all subjected to stringent quality control.



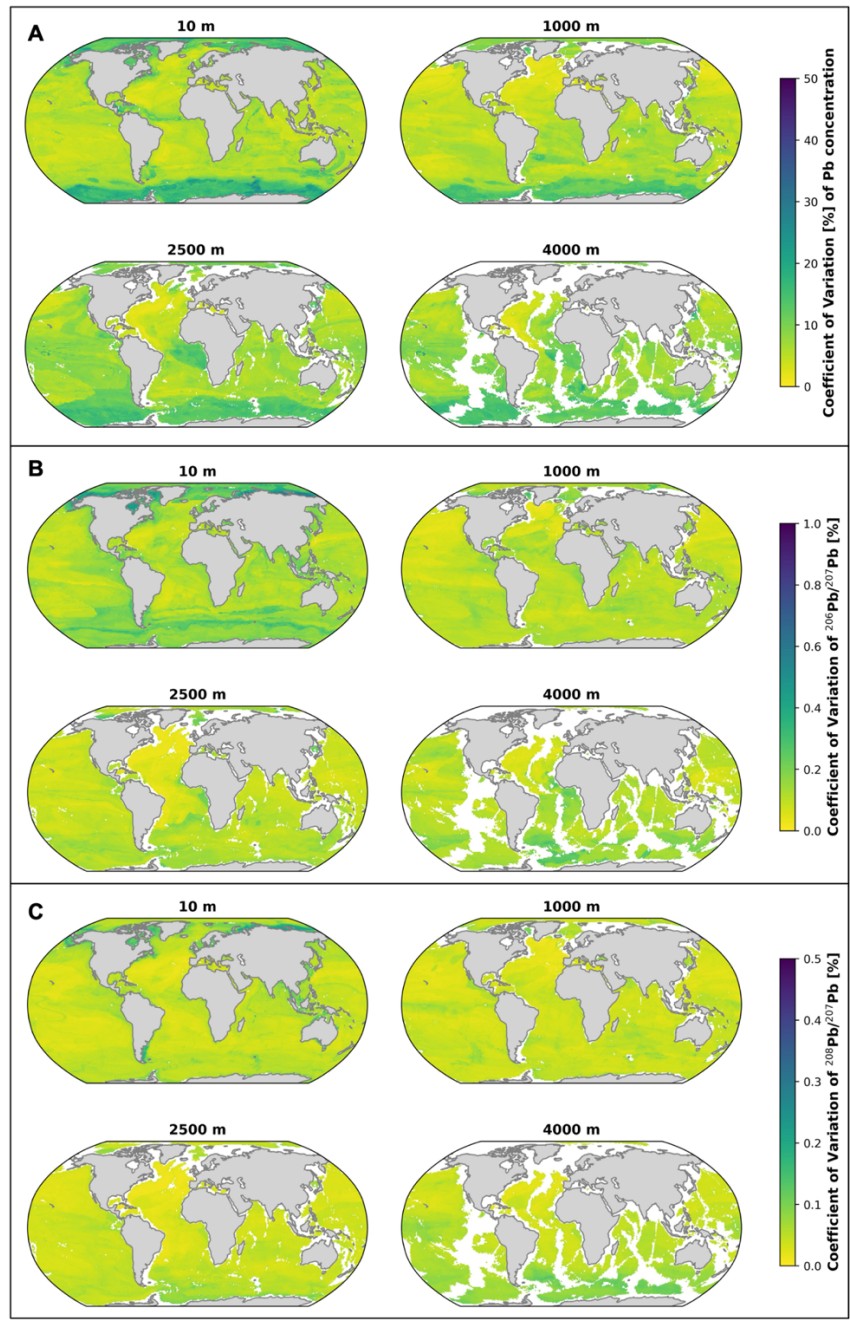

**Figure 8. Coefficient of variation for reconstructed Pb concentration (A), 206Pb/207Pb (B), and 208Pb/207Pb (C) values. As for Figure 4, the different panels represent different depth levels (10 m, 1000 m, 2500 m, 4000 m) and the white patches correspond to seafloor locations. Please note that the colour scales differ for the three panels.**

Lastly, to quantify the uncertainty arising from the random splitting of the data into a training and test set, the best Pb concentration, $^{206}Pb/^{207}Pb$, and $^{208}Pb/^{207}Pb$ model architectures were initiated with 100 different random train-test splits while keeping the hyperparameters constant. The means and standard deviations of the model ensemble predictions for the global maps of Pb concentrations and isotope compositions were used to calculate the coefficients of variation for the mapped Pb

distributions (Fig. 8). The coefficient of variation was calculated as the ratio between the average value of each cell across the 100 different ensemble members and its associated standard deviation. For both Pb concentrations and isotope compositions, the coefficients of variation are highest in the Arctic and Southern Ocean, especially in the surface and intermediate layer (Fig. 8). Both the Arctic and Southern Ocean are characterised by the lowest Pb concentrations observed, therefore any variability in the outputs of the ensemble would cause a larger coefficient of variation as the mean concentrations are low. However, the

spatial variability in observed Pb concentrations in the Southern Ocean is much larger than in the Arctic Ocean. Indeed, high concentrations are found at the surface and in the water column near the Antarctic Peninsula, likely caused by glacial meltwater inputs, while much lower values are visible in the Indian and Australian sectors. Therefore, the different splits between training and test set lead to much more variable model outputs for the Southern Ocean than for its Arctic counterpart. The high latitudes are also the areas with the largest coefficients of variation for $^{206}Pb/^{207}Pb$ and $^{208}Pb/^{207}Pb$, arguably because of the significant

presence of both anthropogenic and natural Pb. Based on the variability of the outputs of the three model ensembles, it can be argued that at-sea sampling efforts should focus on the Southern Ocean to obtain a clearer understanding of the distribution and cycling of Pb and its isotopes in that area, and to reduce the uncertainty associated with our modelled outputs.

**4 Model and data applications**

More than 20 years ago, Henderson & Maier-Reimer (2002) used a general circulation model to investigate the natural marine

cycle of Pb and its isotopes. As these authors did not consider inputs from anthropogenic activities, we here present the first study that provides global maps of the distribution of Pb concentrations, and $^{206}Pb/^{207}Pb$ and $^{208}Pb/^{207}Pb$ ratios.

In addition to providing a detailed demonstration of the applicability of machine learning to marine (isotope) geochemistry, the new models and their outputs can be used to (i) benchmark future levels of pollution, (ii) inform sampling strategies and campaigns, and (iii) compare outputs from different models, either process-based or data-driven. Lead concentration and

isotope composition data from our 12-year climatologies provide useful information on the levels of marine pollution, both in terms of sources and spatial extent. This insight can be used to assess the impact of human activities and any potential environmental policies implemented in the future. Both the mapped distributions and their associated uncertainties can be used by the community to support any process-oriented or large-scale sampling campaigns. In particular, the large uncertainty associated with Pb concentrations in the Southern Ocean and the widespread distribution of low $^{206}Pb/^{207}Pb$ and $^{208}Pb/^{207}Pb$

values at intermediate depth throughout the Southern Hemisphere suggest that these areas should be prioritised in the future. Lastly, the $1° \times 1°$ data products developed will be a useful and robust comparison for any global and regional modelling studies that attempt to parametrise and reproduce the processes driving the biogeochemical cycle of Pb, which still require future work.

## 5 Data and code availability

All scripts generated to build the Pb concentration, $^{206}Pb/^{207}Pb$, and $^{208}Pb/^{207}Pb$ datasets and the corresponding models are available at https://github.com/OlivelliAri/Pb-ML_GEOTRACES. The gridded climatologies of Pb concentrations, $^{206}Pb/^{207}Pb$, and $^{208}Pb/^{207}Pb$, as well as the results of the model ensemble for uncertainty calculations, are available in Olivelli et al. (2024a, https://doi.org/10.5281/zenodo.14261154).

## 6 Conclusion

We successfully generated global climatologies of Pb concentrations, $^{206}Pb/^{207}Pb$, and $^{208}Pb/^{207}Pb$ ratios using the non-linear regression algorithm XGBoost, trained and tested on high-quality Pb data collected as part of the GEOTRACES programme. We found that Pb concentrations are highest in the surface layer of the Indian and North Pacific Ocean and at intermediate depths in the North Atlantic, and lowest in the Arctic Ocean. The $^{206}Pb/^{207}Pb$ and $^{208}Pb/^{207}Pb$ ratios are lowest in the surface Indian Ocean and at intermediate depths in the Southern Hemisphere, and highest throughout the water column in the Arctic

Ocean. Taken together, the distributions of Pb concentrations, $^{206}Pb/^{207}Pb$, and $^{208}Pb/^{207}Pb$ ratios indicate that (i) the surface Indian Ocean has the highest levels of pollution, (ii) pollution from previous decades is sinking in the North Atlantic and Pacific Ocean, and (iii) that a highly anthropogenic fingerprint originating in the Southern Ocean is spreading at intermediate depths throughout the Southern Hemisphere with AAIW. The lower model performance and higher uncertainty associated



with reconstructions of the Southern Ocean suggest that this is a key area to prioritise future sampling campaigns to expand

our understanding of biogeochemical processes that drive the distribution and cycling of Pb and its isotopes in the ocean. More

broadly, our approach demonstrates the applicability of machine learning to marine (isotope) geochemistry, even when data is

scarce and very sparse, as for Pb isotope compositions. Lastly, our model output will provide a useful benchmark for future

levels of pollutions and a valuable comparison for process-based models that aim to improve understanding of the

biogeochemical processes governing the marine cycling of Pb.

**Acknowledgements**

All members of the Data Learning group at Imperial College London are thanked for the useful discussions and suggestions

throughout this study. Rob Middag and Kyyas Seyitmuhammedov are thanked for the information they shared regarding Pb

concentration samples collected near the Antarctic Peninsula on GEOTRACES cruise GPpr08. Francois van Shalkwyk is

thanked for his support with computational resources.

**Funding**

Arianna Olivelli was supported by the Natural Environment Research Council (NERC) (NE/S007415/1). The International

GEOTRACES Programme is possible in part thanks to the support from the U.S. National Science Foundation (Grant OCE-

2140395) to the Scientific Committee on Oceanic Research (SCOR). For the purpose of open access, the author has applied a

Creative Commons Attribution (CC BY) license to any Author Accepted Manuscript version arising.

**Author contribution statement**

Arianna Olivelli devised the project, curated the data, developed the models, led the interpretation and wrote the initial draft

of this manuscript with support from Rossella Arcucci, Mark Rehkämper and Tina van de Flierdt. All authors contributed to

the final version of the manuscript.

**Competing Interests**

The authors declare that they have no conflict of interest.



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
