# Peer review of "Mapping the global distribution of lead and its isotopes in seawater with explainable machine learning"

_Earth System Science Data, 2025_

## Author Comment (AC1)

**Point-by-point response to Reviewer #2.**
**Original review comments are in black, and our responses are in blue.**

General comments

In this manuscript, the authors reconstructed the global distribution of Pb and its isotopes in seawater with machine learning models. I'm not familiar with machine learning at all. So, I make some comments on this manuscript as a sea-going scientist. The models seem unique and provide some interesting points of view for us. For example, black carbon in air is one of the important factors to reconstruct the global Pb distribution in this model. From this result, I realized that the simultaneous determination of black carbon and Pb isotopes in aerosols is a good topic in the field studies. I think this manuscript is challenging but has possibility to give us some insight on the marine biogeochemical cycles of Pb.

We thank this anonymous reviewer for the positive and constructive feedback. We provide below detailed responses to all comments made by the reviewer.

Specific comments

Line 310, "km/m$^3$": Is this unit correct?

Thanks for spotting this typo. The unit has been corrected to "kg/m$^3$" (L311).

Line 322 - 325: Is this meaningful description?

We revised the whole paragraph and omitted the sentence in question from the revised version of the manuscript.

Line 365 – 366, "further studies of the Pb isotope compositions of dust and black carbon sources to the atmosphere": I could not find so many references for these topics, but at least one reference concerning the topics (e.g. Nizam et al., 2020) should be mentioned.

We thank the reviewer for the suggested reference, which has now been added to the text (L364).

Line 370 – 372, "Moreover, black carbon is not only emitted due to fossil fuel consumption, but also from forest fires and coal combustion, which might have substantially different Pb isotope signatures compared to other industrial and urban sources.": Recently, sources of black carbon in aerosol were discussed from the radiocarbon measurements (Gustafsson et al., 2009; Li et al., 2016). Considering these references, the authors should discuss the sources of the black carbon at this moment.

We expanded our discussion on black carbon sources and included the references suggested by the reviewer in the revised manuscript (L368-374).

Line 385 – 386, "The only exception to this trend is the North Atlantic Ocean, where mapped Pb concentrations at 1000 m and 2500 m are higher than at 10 m.": From the observational studies, subsurface maxima of Pb were reported in the North Pacific (Wu et al., 2010; Zurbrick et al., 2017; Zheng et al., 2019; Chan et al., 2024; Jiang et al., 2025). Were these features found in this model?

Yes, they were also observed in the model but at different depths than the intervals presented in the figures (between 70 and 500 m depending on location), hence they were previously not discussed. However, for clarity, we have modified the text between L388 and L392, and this now mentions the subsurface maxima in the North Pacific in agreement with observations.

Line 400 - 403: In the Northwest Pacific, the subduction and ventilation process of the North Pacific mode waters and NPIW were considered to elevate Pb concentrations in the subsurface layers (Jiang et al., 2021). Since numbers of data are relatively small in the Northwest Pacific, this feature might not be captured in this model.

Yes, in agreement with the answer above, we also see subsurface maxima in the Northwest Pacific. The sentences between L404 and L409 were modified to include this point in the discussion.

References

Chan, C., L. Zheng, Y. Sohrin (2024). The behavior of aluminium, manganese, iron, cobalt, and lead in the subarctic Pacific Ocean: boundary scavenging and temporal changes. Journal of Oceanography 80, 99 – 115.

Gustafsson, Ö., M. Kruså, Z. Zencak, R. J. Sheesley, L. Granat, E. Engström, P. S. Praveen, P. S. P. Rao, C. Leck, H. Rodhe (2009). Brown Clouds over South Asia: Biomass or Fossil Fuel Combustion? Science 323, 495 – 498.

Jiang, S., J. Zhang, H. Zhou, Y. Xue, W. Zheng (2021). Concentration of dissolved lead in the upper Northwestern Pacific Ocean. Chemical Geology 577, 120275.

Jiang, S., N. Lanning, E. Boyle, J. Fitzsimmons, J. Ramezani, A. G. Wang, J. Zhang (2025). Meridional central Pacific Ocean depth section for Pb and Pb isotopes (GEOTRACES GP15, 152°W, 56°N to 20°S) including shipboard aerosols. Journal of Geophysical Research: Oceans 130, e2024JC021674.

Li, C., C. Bosch, S. Kang, A. Andersson, P. Chen, Q. Zhang, Z. Cong, B. Chen, D. Qin, O. Gustafsson (2016). Sources of black carbon to the Himalayan–Tibetan Plateau glaciers. Nature Communications 7, 12574.

Nizam, S., I. S. Sen, V. Vinoj, V. Galy, D. Selby, M. F. Azam, S. K. Pandey, R. A. Creaser, A. K. Agarwal, A. P. Singh, M. Bizimis (2020). Biomass-Derived Provenance Dominates Glacial Surface Organic Carbon in the Western Himalaya. Environmental Science and Technology 54, 8612 − 8621.

Wu, J., R. Rember, M. Jin, E. A. Boyle, A. R. Flegal (2010). Isotopic evidence for the source of lead in the North Pacific abyssal water. Geochimica et Cosmochimica Acta 75, 460 – 468.

Zheng, L., T. Minami, W. Konagaya, C. Chan, M. Tsujisaka, S. Takano, K. Norisuye, Y. Sohrin (2019). Distinct basin-scale-distributions of aluminum, manganese, cobalt, and lead in the North Pacific Ocean. Geochimica et Cosmochimica Acta 254, 102-121.

Zurbrick, C. M., C. Gallon, A. R. Flegal (2017). Historic and industrial lead within the Northwest Pacific Ocean evidenced by lead isotopes in seawater. Environmental Science and Technology 51, 1203 – 1212.

---

## Author Comment (AC2)

**Point-by-point response to Reviewer #1 – Edward Boyle.**
**Original review comments are in black, and our responses are in blue.**

Review of "Mapping the global distribution of lead and its isotopes in seawater with explainable machine learning" by Olivelli et al. (essd 2025-17).

Although this reviewer knows something about Pb and Pb isotopes in the ocean, I do not know anything about AI models. So, my comments are addressed strictly to what the paper says about Pb and Pb isotopes in the ocean without any critical assessment of the AI model.

My bottom lines on this effort to see if AI is useful in understanding Pb in the ocean is given by my inferred (from this manuscript) answers to two questions:

(1) *Is AI a useful way to summarize global 3D patterns of Pb and Pb isotopes in the ocean given very limited data?* YES – the maps correspond decently with what is known for sure about Pb in the ocean, both from data and from understood processes. There are some implied features that have no observational basis (e.g. the high surface Pb values in the tropical eastern Pacific), but those can be corrected as future Pb data is published.

(2) *Has this AI effort increased our understanding of Pb in the ocean?* NO – the conclusions as stated in the abstract and conclusions section were already known from the raw data (and are stated as such in publications), I don't see any advance in our understanding of Pb in the ocean. However, it does represent an advance to the question "Of what utility can AI have now in studying Pb in the ocean?"

Overall, I support publication after relatively minor changes.

We thank Edward Boyle for the constructive review of our manuscript. We appreciate his feedback and provide below detailed responses to all comments.

Specific comments on sections:
Line 22: "Our model outputs show that…" Baloney, the features that follow were already evident in the raw data. The model is just mimicking the data for the listed features.

We have rephrased the sentence to "In line with observations, our model outputs show that …" (L22).

Line 38: "did not allow for successful measurement of seawater Pb concentration measurements until 1963 (refs. Tatsumoto and Petterson, 1963)". I don't agree that the 1963 data set was correct, and therefore not the first. They are high compared with the coral measurements of Desenfant et al. (2006, Coral Reefs 25:473) and Kelly et al. (2009, EPSL 283:93). The first successful Pb measurements in the Atlantic are the 1979 data of Schaule and Patterson (1983, in Trace Elements in Seawater, eds. C.S. Wong et al.).

We appreciate the reviewer's insight on the Pb concentration measurements of Tatsumoto and Patterson (1963) and agree that the values obtained by the authors in 1963 for the samples

collected in the North Atlantic are higher than those by Kelly et al. (2009) and Desenfant et al. (2006). The sentence in the manuscript was based on a recent Nature article by Jerome Nriagu (2023), who reported that the 1963 measurements opened the way for marine trace metal isotope geochemistry. On reflection, however, we agree with the reviewer that we should refer here to the first successful seawater measurements in the Atlantic and Pacific Oceans (Schaule and Patterson, 1981 and 1983). We amended the sentence in the manuscript accordingly.

Lines 115-116: "Dust was found to be an important source of natural Pb to the surface ocean in several regions, including the … southern Indian Ocean (Lee et al., 2015)". That is not true. Lee et al. did see high $^{206}Pb/^{207}Pb$ ratios at 62°S *deep waters* consistent with crustal Pb, but they did not attribute it to continental dust because there is very little crustal aerosol dust in the Southern Ocean today. It seems more likely that this feature is caused by glacial erosion from the Antarctic continent.

Our statement was based on Section 4.1 in Lee et al. (2015), which discusses the origin of Pb throughout the water column at 62°S. While the authors did not explicitly state that dust was the sole source of Pb to the sampling station, they considered the potential role of dust as a source of natural Pb to the Southern Ocean and did not rule it out. As the reviewer is a co-author of the study by Lee et al. (2015) and pointed out that our interpretation of the article seems incorrect, we removed the reference from our manuscript.

Lines 270- : re the role of oxygen as a fitting device: I don't think there is any reasonable mechanistic driver for such a correlation, with no obvious indication of "Pb regeneration from sinking organic matter" in the observations. However, for example in the Northeast Atlantic, the water with the lowest $O_2$ has an $SF_6$ age of about 40 years, which was the period of maximum Atlantic region leaded gasoline utilization. So, the correlation is simply based on the aging of the water mass ($O_2$ decrease) and the coincidence of that maximum with the period of highest Pb emissions.

Similarly with regard to back carbon as a fitting parameter, yes, Pb and black carbon are both anthropogenic in origin and it isn't surprising if they often show similar spatial sources (where are people, cars and industrial activities), however, they sometimes should not be correlated (e.g., it isn't clear the tropical forest burning is much of a Pb source but it is for black carbon).

We thank the reviewer for the comment regarding the choice of oxygen as a predictor for the model and would like to clarify that we did not include it based on the assumption of "Pb regeneration from sinking organic matter", as we agree that this is not visible from observations. Rather, our rationale was to include it as the distribution of dissolved $O_2$ provides complementary information to that of the other major oceanographic features considered.

The suggested explanation of water mass aging ($O_2$ decrease) and corresponding historical Pb emissions seems reasonable for the North Atlantic. However, the SHAP explanation of our model indicates that higher values of dissolved $O_2$ (in areas with low nitrate concentrations) lead to higher $^{206}Pb/^{207}Pb$ ratios on a *global scale*, and vice versa, as discussed in L304-307. As the Pb isotope compositions of anthropogenic and natural sources of Pb vary on a regional scale, and $O_2$ concentrations in the North Atlantic are generally higher than in other basins (excluding the Arctic), we believe that the relationship observed on a global scale between $O_2$ and predicted $^{206}Pb/^{207}Pb$ values takes into account the regional variations of Pb sources and dissolved $O_2$ concentrations at the surface and throughout the water column, and does not solely reflect water mass age. No changes were made.

Regarding black carbon, we agree with the reviewer's comment on the correlation between industrial Pb and black carbon sources and expected lack thereof for sources such as forest fires. We now discuss this in more detail in lines 360-375, and, upon reviewer #2's suggestion, have added additional references on current sources of black carbon (L362, L366-372).

Lines 422- : "However, historically Pb isotopes have been regarded as tracers of water mass movements and ventilation (refs. Veron et al., 1998, 1999)" – This is only true for the North Atlantic, and only because (in John Edmond's words) "the Atlantic is a bowling alley" with strong and rapid lateral water mass movements adjacent to recognizable sources. This statement is not true for the Pacific.

We agree with the reviewer and have clarified this point in L427, with the sentence now reading "However, historically Pb isotopes have been regarded as tracers of water mass movements and ventilation in the North Atlantic Ocean (e.g. Véron et al., 1998, 1999)".